# Variational Bayesian Decision-making for Continuous Utilities

**Tomasz Kuśmierczyk**     **Joseph Sakaya**     **Arto Klami**
Helsinki Institute for Information Technology HIIT
Department of Computer Science, University of Helsinki
{tomasz.kusmierczyk,joseph.sakaya,arto.klami}@helsinki.fi

## Abstract

Bayesian decision theory outlines a rigorous framework for making optimal decisions based on maximizing expected utility over a model posterior. However, practitioners often do not have access to the full posterior and resort to approximate inference strategies. In such cases, taking the eventual decision-making task into account while performing the inference allows for calibrating the posterior approximation to maximize the utility. We present an automatic pipeline that co-opts continuous utilities into variational inference algorithms to account for decision-making. We provide practical strategies for approximating and maximizing the gain, and empirically demonstrate consistent improvement when calibrating approximations for specific utilities.

## 1  Introduction

A considerable proportion of research on Bayesian machine learning concerns itself with the task of *inference*, developing techniques for an efficient and accurate approximation of the posterior distribution $p(\theta|\mathcal{D})$ of the model parameters $\theta$ conditional on observed data $\mathcal{D}$. However, in most cases, this is not the end goal in itself. Instead, we eventually want to solve a *decision problem* of some kind and merely use the posterior as a summary of the information provided by the data and the modeling assumptions. For example, we may want to decide to automatically shut down a process to avoid costs associated with its potential failure, and do not care about the exact posterior as long as we can make good decisions that still account for our uncertainty of the parameters.

Focusing on inference is justified by *Bayesian decision theory* [2] formalizing the notion that the posterior is sufficient for making optimal decisions. This is achieved by selecting decisions that maximize the expected utility, computed by integrating over the posterior. The theory, however, only applies when integrating over the *true* posterior which can be computed only for simple models. With approximate posteriors it is no longer optimal to separate inference from decision-making. Standard approximation algorithms try to represent the full posterior accurately, yet lack guarantees for high accuracy for parameter regions that are critical for decision-making. This holds for both distributional techniques, such as variational approximation [4] and expectation propagation [10, 18], as well as Markov chain Monte Carlo (MCMC) – even though the latter are asymptotically exact, a finite set of samples is still an approximation and it is often difficult to sample from the correct distribution.

*Loss-calibrated inference* refers to techniques that adapt the inference process to better capture the posterior regions relevant to the decision-making task. First proposed by Lacoste-Julien et al. [16] in the context of variational approximation, the principle has been used also for calibrating MCMC [1], and recently for Bayesian neural networks [7]. The core idea of loss calibration is to maximize the expected utility computed over the approximating distribution, instead of maximizing the approximation accuracy, while still retaining a reasonable posterior approximation. Figure 1 demonstrates how the calibration process shifts an otherwise sub-optimal approximation to improve the decisions,

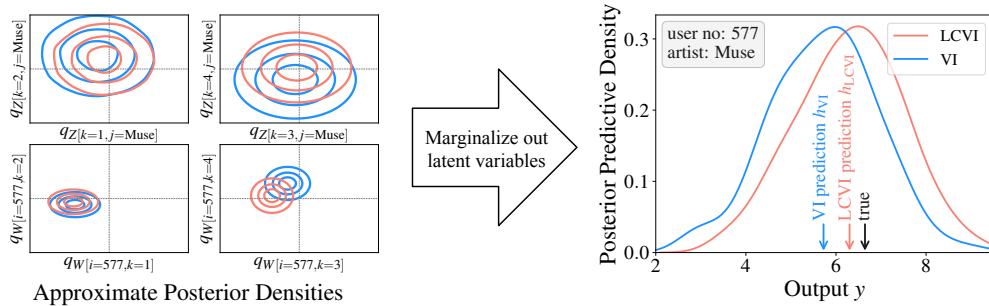

Figure 1: Loss-calibration (red) modifies the posterior approximation (left) so that Bayes optimal decisions for the predictive distribution (right) are better in terms of a user-defined loss, here squared error, while still characterizing the posterior almost as well as the standard variational approximation (blue). See Section 6.2 for detailed description of the experiment.

but still represents the uncertainty over the parameter space. That is, we are merely fine-tuning – calibrating – the approximation instead of solely optimizing for the decision.

Previous work on calibrating variational approximations only deals with classification problems [7, 16] making discrete decision amongst finitely many classes. This allows algorithms based on explicit enumeration and summation of alternative decisions, which are inapplicable for continuous spaces. Lack of tools for continuous utilities has thus far ruled out, for example, calibration for regression problems. We provide these tools. We analyse the degree of calibration under linear transformations of utility, and describe how efficient calibration can be carried out also when the user characterises the relative quality of the decisions with *losses* instead of utilities. To cope with the challenges imposed by moving from discrete to continuous output spaces, we replace the enumeration over possible choices by nested Monte Carlo integration combined with double reparameterization technique, and provide algorithms for learning optimal decisions for a flexible choice of utilities. We demonstrate the technique in predictive machine learning tasks on the eight schools model [9, 29] and probabilistic matrix factorization on media consumption data.

## 2 Background

### 2.1 Bayesian decision theory

Bayesian decision theory [2, 24] is the axiomatic formalization of decision-making under uncertainty. Given a posterior distribution $p(\theta|\mathcal{D})$ of a parametric model conditioned on data $\mathcal{D}$, we desire to make optimal *decisions* $h$. The value of individual decisions depends on the *utility* $\tilde{u}(\theta, h) \geq 0$ that is a function of both: (1) the state of the world $\theta$ and (2) the decision $h$. The optimal decisions $h_p$ maximize the *gain* (=the expected utility)

$$\mathcal{G}_u(h) = \int p(\theta|\mathcal{D})\tilde{u}(\theta, h)d\theta.$$

An equivalent formulation is obtained by evaluating individual decisions by a *loss* function $\tilde{\ell}(\theta, h)$ and solving for optimal decisions by minimizing the *risk* $\mathcal{R}_l(h) = \int p(\theta|\mathcal{D})\tilde{\ell}(\theta, h)d\theta$.

Even though some decision problems operate directly on model parameters $\theta$, it is more typical to make decisions regarding predictions $y \sim p(y|\mathcal{D})$. For such problems the utility is expressed as $u(y, h)$, which together with the model induces the utility $\tilde{u}(\theta, h) = \int p(y|\theta, \mathcal{D})u(y, h)dy$, where we use the notation $p(y|\theta, \mathcal{D})$ to indicate the prediction may depend on some covariates in $\mathcal{D}$. This complicates computation because evaluating the gain requires nested integration over $p(\theta|\mathcal{D})$ and $p(y|\theta, \mathcal{D})$. In the remainder of the paper we focus on this more challenging family of decisions, and always use $\tilde{u}(\theta, h)$ to denote the expected utility induced by the predictive utility $u(y, h)$.

## 2.2 Variational inference

Variational inference approximates the posterior $p(\theta|\mathcal{D})$ with a proxy distribution $q_\lambda(\theta)$ parameterized by $\lambda$, typically by maximizing a lower bound $\mathcal{L}_{\text{VI}}(\lambda)$ for the marginal log-likelihood

$$\log p(\mathcal{D}) = \log \int q_\lambda(\theta) \frac{p(\mathcal{D}, \theta)}{q_\lambda(\theta)}\, d\theta \geq \int q_\lambda(\theta) \log \frac{p(\mathcal{D}, \theta)}{q_\lambda(\theta)}\, d\theta =: \mathcal{L}_{\text{VI}}(\lambda).$$

Traditional methods use coordinate ascent updates, mean-field approximations, and conjugate priors for computational tractability [4]. Recently, several gradient-based optimization algorithms [5, 22, 28] have made variational inference feasible for non-conjugate models and richer approximation families, using gradient-based optimization of Monte Carlo estimates of the bound.

The most efficient techniques use reparameterization to compute the gradients $\nabla_\lambda \mathcal{L}_{\text{VI}}(\lambda) = \nabla_\lambda \mathbb{E}_{q_\lambda}(\theta)[\log p(\mathcal{D}, \theta) - \log q_\lambda(\theta)]$, by rewriting the distribution $q_\lambda(\theta)$ using a differentiable transformation $\theta = f(\epsilon, \lambda)$ of an underlying, parameter-free standard distribution $q_0(\epsilon)$ [28]. We can then use Monte Carlo integration over $q_0(\epsilon)$ for evaluating the expectations, yet the value depends on $\theta$ and hence we can propagate gradients through $f(\cdot)$ for learning. The reparameterization can be carried out either explicitly [20, 25] or implicitly [8]; the latter strategy makes reparameterization possible for almost any distribution. Our derivations and experiments are on simple parametric approximations, but we note that the loss calibration elements can be combined with wide range of recent advances in variational inference, such as generalized VI [14], boosting VI [11, 17], more efficient structural approximations [12], and normalizing flows for flexible approximations [23].

## 2.3 Loss-calibrated variational inference

The idea of calibrating a variational approximation was proposed by Lacoste-Julien et al. [16], based on lower bounding the logarithmic gain using Jensen's inequality as

$$\log \mathcal{G}_u(h) = \log \int \frac{q_\lambda(\theta)}{q_\lambda(\theta)} p(\theta|\mathcal{D}) \tilde{u}(\theta, h)\, d\theta \geq -\text{KL}(q, p) + \underbrace{\int q_\lambda(\theta) \log \tilde{u}(\theta, h) d\theta}_{\mathbb{U}(\lambda, h) \text{ - utility-dependent term}}.$$

The bound consists of two terms. The first term, negative Kullback-Leibler divergence between the approximation and the posterior, can be further replaced by a lower bound for the marginal likelihood [4] analogous to standard variational approximation to provide the final bound $\mathcal{L}(\lambda, h) := \text{ELBO}(\lambda) + \mathbb{U}(\lambda, h) \leq \log \mathcal{G}_u(h)$. The second term accounts for decision making. It is independent of the observed $y$ and only depends on the current approximation $q_\lambda(\theta)$, favoring approximations that optimize the utility. For efficient optimization the bound for multiple predictions can be written as sum over individual data instances as

$$\mathcal{L}(\lambda, \{h\}) = \sum_{i \in [\mathcal{D}]} \left( \text{ELBO}_i(\lambda) + \mathbb{U}(\lambda, h_i) \right) \leq \log \mathcal{G}_u(\{h\}), \tag{1}$$

where $\text{ELBO}_i$ accounts for an individual data point $y_i$ for which the hypothesis is $h_i$. This holds for predictive models that assume i.i.d. predictions and additive log-gains, which captures most practical scenarios and allows for making decisions $h_i$ separately for individual data points. For clarity, we drop the subscript $i$ in $h_i$ during derivations.

For optimizing the bound, Lacoste-Julien et al. [16] derived an EM algorithm that alternates between learning optimal $\lambda$ and selecting optimal decisions:

$$\text{E-step: } \lambda := \arg\max_\lambda \mathcal{L}(\lambda, \{h\}), \qquad \text{M-step: } \{h\} := \arg\max_{\{h\}} \mathcal{L}(\lambda, \{h\}) = \arg\max_{\{h\}} \mathbb{U}(\lambda, \{h\}).$$

However, they used closed-form analytic updates for $\lambda$, for which incorporating the utility-dependent term is difficult, and only demonstrated the principle in classification problems with discrete $h$. Cobb et al. [7] derived a loss-calibrated variational lower bound for Bayesian neural networks with discrete decisions $h$. Unlike Lacoste-Julien et al. [16], however, their updates for $\lambda$ are gradient-based and applicable to generic utility-dependent terms as long as the decisions are discrete.

## 3 Loss calibration for continuous utilities

Handling continuous decisions requires both more careful treatment of utilities and losses, described below, and new algorithms for optimizing the bound, provided in Section 4.

## 3.1 The calibration effect

Optimal decisions are invariant to linear transformations of the utility, so that $\arg\max_h \mathcal{G}_u(h) = \arg\max_h \mathcal{G}_{u'}(h)$ for $u'(y,h) = \alpha \cdot u(y,h) + \beta$ for $\alpha > 0$ [2]. However, this does not hold for the loss-calibration procedure. Instead, translating the utilities with $\beta$ influences the degree of calibration. To see this we assume (for notational simplicity) $\inf_{y,h} u(y,h) = 0$ and write $\mathbb{U}$ corresponding to $u'(y,h)$ as

$$\mathbb{E}_q \left[ \log \left( \beta + \alpha \int p(y|\theta,\mathcal{D})u(y,h)dy \right) \right] = \underbrace{\mathbb{E}_q \left[ \log \left( 1 + \frac{\alpha}{\beta} \int p(y|\theta,\mathcal{D})u(y,h)dy \right) \right]}_{\text{expectation term}} + \log\beta,$$

where the equality holds for any $\beta > 0$ (negative values would lead to negative utilities). Since $\log\beta$ is constant w.r.t variational parameters $\lambda$, only the expectation term is relevant for optimization. However, its impact (=magnitude) relative to ELBO depends on the ratio $\frac{\alpha}{\beta}$. In particular, as $\beta \to \infty$ (for any fixed $\alpha$) the expectation term converges to $0$ removing the calibration effect completely. In contrast, pushing $\beta \to 0$ maximizes the effect of calibration by maximizing the magnitude of $\mathbb{U}$. Hence, maximal calibration is achieved when $\beta = 0$, i.e., when $\inf_{y,h} u'(y,h) = 0$. Finally, for $\beta = 0$ the scaling constant $\alpha$ can be taken out of the expectation and hence has no effect on optimal solution.

In summary, the calibration effect is maximized by using utilities with zero infimum, and for such utilities the procedure is scale invariant. The procedure is valid also when this does not hold, but the calibration effect diminishes and depends on the scaling in an unknown manner.

## 3.2 Utilities and losses

As stated in Section 2.1, decision problems can be formulated in terms of maximizing gain defined by a utility $u(y,h) \geq 0$, or in terms of minimizing risk defined by a loss $\ell(y,h) \geq 0$. The calibration procedure above is provided for the gain, since both the bound for the marginal likelihood and the bound for the gain need to be in the same direction in Eq. (1). To calibrate for user-defined loss (which tends to be more common in practical applications), we need to convert the loss into a utility. Unfortunately, the way this is carried out influences the final risk evaluated using the original loss.

Only linear transformations retain the optimal decisions, and the simplest one providing non-negative utilities is $u(y,h) = M - \ell(y,h)$ where $M \geq \sup_{y,h} \ell(y,h)$ [2], with equality providing optimal calibration as explained above. However, we cannot evaluate $M = \sup_{y,h} \ell(y,h)$ for continuous unbounded losses. Furthermore, this value may be excessively large due to outliers, so that $M \gg \ell(y,h)$ for almost all instances, effectively removing the calibration even if we knew how to find the optimal value. As a remedy, we propose two practical strategies to calibrate for continuous unbounded losses, both based on the intuitive idea of bringing the utilities close to zero for the losses we expect to see in practice.

**Robust maximum** For $u(y,h) = M - \ell(y,h)$, any value of $\ell(y,h) > M$ may lead to $\tilde{u}(\theta,h) < 0$ and hence to negative input to $\log$ in $\mathbb{U}$. However, this problem disappears if we linearize the logarithm in $\mathbb{U}$ around $M$ similar to [16]. Using Taylor's expansion

$$\log\tilde{u}(\theta,h) = \log(M - \tilde{\ell}(\theta,h)) = \log M - \frac{\tilde{\ell}(\theta,h)}{M} + \mathcal{O}\left( \frac{\tilde{\ell}(\theta,h)^2}{M^2} \right),$$

and dropping the error term, $\log\tilde{u}(\theta,h) \approx \log M - \frac{\tilde{\ell}(\theta,h)}{M}$ and the utility-dependent term $\mathbb{E}_q[\log\tilde{u}(\theta,h)]$ can be re-expressed as $\mathbb{E}_q\left[ \log M - \frac{\tilde{\ell}(\theta,h)}{M} \right] = \log M - \frac{1}{M}\mathbb{E}_q\left[ \tilde{\ell}(\theta,h) \right]$. We can ignore $\log M$ as it is constant with respect to the decisions $h$ and variational parameters $\lambda$. Now that $\tilde{u}(\theta,h)$ no longer appears inside a log, we can use also $M$ that is not a strict upper bound, but instead a robust estimator for maximum that excludes the tail of the loss distribution. We propose

$$u(y,h) = M_q - \ell(y,h), \tag{2}$$

where $M_q$ is the $q$th quantile (e.g. 90%) of the expected loss distribution. To obtain $M_q$ we first run standard VI for sufficiently many iterations (until the losses converge), and then compute the loss for

every training instance. We then sort the resulting losses and set $M_q$ to match the desired quantile of this empirical distribution of losses. In many cases, $M_q$ is considerably smaller than $\sup_{y,h} l(y, h)$, which increases the calibration effect.

**Non-linear loss transformation**  The other alternative is to use transformations that guarantee non-negative utilities by mapping losses into positive values. A practical example is

$$u(y, h) = e^{-\gamma \ell(y,h)}, \tag{3}$$

where the rate parameter $\gamma$ can be related to the quantiles of the loss distribution as $\gamma = M_q^{-1}$, by solving for a value for which linearization of the utility at $\ell(y, h) = 0$ would be zero for $\ell(y, h) = M_q$.

## 4   Algorithms for calibrating variational inference

For practical computation we need to optimize Eq. (1) w.r.t. both $\lambda$ and $\{h\}$ in a model-independent manner, which can be carried out using techniques described next.

### 4.1   Monte Carlo approximation of $\mathbb{U}$

The first practical challenge concerns evaluation and optimization of $\mathbb{U} = \int q_\lambda(\theta) \log \tilde{u}(\theta, h) d\theta = \int q_\lambda(\theta) \log \int p(y|\theta, \mathcal{D}) u(y, h) dy d\theta$. Since we already reparameterized $\theta$ for optimization of ELBO, we can as well approximate the outer expectation as

$$\mathbb{U}(\lambda, h) \approx \frac{1}{S_\theta} \sum_{\theta \sim q_\lambda(\theta)} \log \tilde{u}(\theta, h) = \frac{1}{S_\theta} \sum_{\epsilon \sim q_0} \log \tilde{u}(f(\epsilon, \lambda), h), \tag{4}$$

where $q_0$ is the zero-parameter distribution, $f$ transforms samples from $q_0(\epsilon)$ into samples from $q_\lambda(\theta)$, and $S_\theta$ is a number of samples for Monte Carlo integration. For discrete outputs the inner expectation computing $\tilde{u}(\theta, h)$ becomes a sum over possible values $\mathcal{Y}$, which makes $\mathbb{U}$ straightforward to optimize both w.r.t. $\lambda$ (via gradient ascent) and $h$ (by enumeration). For continuous $y$, however, the integral remains as the main challenge in developing efficient algorithms.

We address this challenge by a *double reparametrization* scheme. Besides reparameterizing the approximation $q_\lambda(\theta)$, we reparameterize also the predictive likelihood $p(y|\theta, \mathcal{D})$, what was made possible for most densities by implicit reparameterization gradients [8]. This enables approximating the inner integral with MC samples as

$$\tilde{u}(\theta, h) \approx \frac{1}{S_y} \sum_{\delta \sim p_0} u(g(\delta, \theta, \mathcal{D}), h), \tag{5}$$

while preserving differentiability w.r.t. both $\lambda$ and $h$. Here $\delta$ denotes samples from parameter-free distribution $p_0$ used to simulate samples $y \sim p(y|f(\epsilon, \lambda), \mathcal{D})$ via the transformation $g(\cdot)$. Similar derivation for approximation of the utility-dependent term exists for discrete decisions [7]. This however, does not require the double reparameterization scheme proposed here.

For evaluating $\mathbb{U}$ we use a *naive* estimator that simply plugs (5) in place of $\tilde{u}(f(.), h)$ into (4):

$$\mathbb{U}(\lambda, h) \approx \frac{1}{S_\theta} \sum_{\epsilon \sim q_0} \log \left( \frac{1}{S_y} \sum_{\delta \sim p_0} u(g(\delta, f(\epsilon, \lambda), \mathcal{D}), h) \right). \tag{6}$$

Even though this estimator is slightly biased, it works well in practice. The bias could be reduced, for example, by employing the Taylor expansion of $\mathbb{E}_p[\log u]$ in a manner similar to [27], or removed by bounding $\mathbb{U}$ with Jensen's inequality as $\mathbb{U}(\lambda, h) \geq \int q_\lambda(\theta) \int p(y|\theta, \mathcal{D}) \log u(y, h) dy d\theta$, but such estimators display numerical instability for utilities close to zero and are not useful in practice.

Finally, we note that for linearized utility-dependent term, for problems defined in terms of losses, we can directly use the simple unbiased estimator

$$\mathbb{U}(\lambda, h) \approx -\frac{1}{M S_\theta S_y} \sum_{\epsilon \sim q_0} \sum_{\delta \sim p_0} \ell(g(\delta, f(\epsilon, \lambda), \mathcal{D}), h). \tag{7}$$

We note that, however useful in practice, the linearization of $\mathbb{U}$ may violate the bound in Eq. (1).

Table 1: Losses and their closed-form Bayes estimators that minimize their posterior expected value.

| Loss | Expression | Bayes estimator |
|------|------------|-----------------|
| Squared | $(h - y)^2$ | $\mathbb{E}_p[y]$ |
| LinEx | $e^{c(h-y)} - c(h - y) - 1$ | $-\frac{1}{c} \log \int e^{-cy} p(y) dy$ |
| Absolute | $\lvert h - y \rvert$ | $\text{median}_p[y]$ |
| Tilted | $\begin{cases} q \cdot \lvert h - y \rvert & y \geq h \\ (1 - q) \cdot \lvert h - y \rvert & y < h \end{cases}$ | $q\text{-percentile}_p[y]$ |

## 4.2 Optimization

Gradient-based optimization w.r.t. $\lambda$ is easy with automatic differentiation, and hence we focus here on the optimization w.r.t. $h$, first in M-step of EM algorithm and then jointly with $\lambda$.

**Closed-form optimal decision** If the utility is expressed in terms of a loss and we use the linearized estimator (7), the optimal $h$ corresponds to the *Bayes estimator* for the loss and can often be computed in closed form as a statistic of the posterior predictive distribution $p(y|\mathcal{D})$. Some examples are listed in Table 1. However, we typically do not have the predictive distribution in closed form, and hence, the statistics are estimated by sampling from the predictive distribution.

**Numerical optimization** When no closed-form solution for optimal decision is available, we need to numerically optimize a Monte Carlo estimate of $\mathbb{U}$. One could consider parallel execution of multiple one-dimensional solvers, but a more practical approach is to jointly optimize for all $\{h\}$ using an objective aggregated over data (mini-)batch $\mathcal{D}$: $\arg\max_{\{h\}} \sum_{i \in [\mathcal{D}]} \mathbb{U}(\lambda, h_i)$ allowing for use of standard optimization routines with efficient implementations. Gradient-based optimization w.r.t. $\{h\}$ is made easier by the observation that the expectation $\tilde{u}(\theta, h)$ (also when approximated by sampling) tends to be relatively smooth even when the underlying utility $u(y, h)$ is not [26].

**Joint optimization of decisions and approximation parameters** For numerical optimization, we alternatively propose to think of $\{h\}$ as additional parameters and jointly optimize for (1) w.r.t. both $\lambda$ and $\{h\}$ using $\nabla \mathcal{L} = [\nabla_\lambda \text{ELBO}, 0, \ldots, 0]^T + [\nabla_\lambda \mathbb{U}, \frac{\partial \mathbb{U}}{\partial h_1}, \ldots, \frac{\partial \mathbb{U}}{\partial h_{|\mathcal{D}|}}]^T$. This has the advantage of not requiring full numerical optimization of $\{h\}$ for every step, but comes with additional memory overhead of storing $\{h\}$ for the whole data set even when using mini-batches for optimization.

## 5 Automatic VI for decision-making

Building on the above ideas, we provide a practical procedure for calibrating variational approximations for continuous utilities. A problem specification consist of (a) a differentiable model $p(y, \theta)$, typically specified using a probabilistic programming language, along with training data $\mathcal{D}$, (b) a utility (or loss) expressing the quality of decisions, and (c) an approximating family $q_\lambda(\theta)$.

For problems defined in terms of $u(y, h)$, the utilities should be scaled so that $\inf_{\theta, y, h} u(y, h) = 0$, to achieve maximal calibration. For problems defined in terms of loss $\ell(y, h)$, we need to first transform the loss into a utility. For unbounded losses this should be done with a transformation such that utilities corresponding to losses below a suitable upper quantile (typically between 50% and 95%) of the expected loss distribution (or the empirical loss distribution of uncalibrated VI) remain positive. Given the quantile, we can either linearize the utility and use (2) or use the exponential transformation (3), neither of which has no tuning parameters besides the quantile.

To maximize the bound (1) we use joint gradient-based optimization over $\lambda$ and $\{h\}$, reparameterization for $\theta$ and $y$, and (6) or (7) for evaluating $\mathbb{U}$ with sufficiently many samples (*e.g.*, $S_\theta S_y \approx 300$). For problems specified in terms of losses with known Bayes estimators (Table 1), one can also use EM algorithm where optimal decisions are determined by statistics of posterior predictive samples.

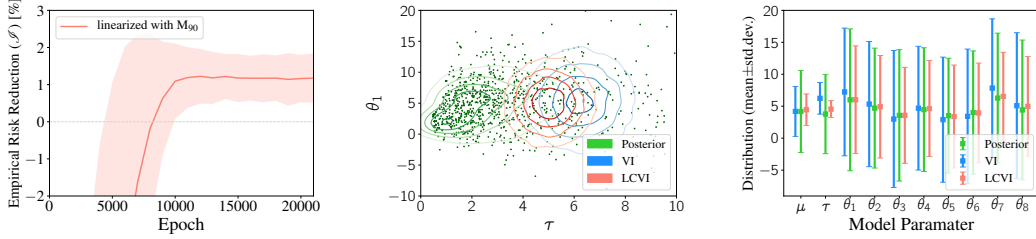

Figure 2: Eight-schools model calibrated for tilted loss ($q = 0.2$). LCVI consistently reduces the risk (left) while shifting the posterior approximation (middle) and compressing marginal densities (right).

## 6 Experiments

We first demonstrate how the calibration affects the posterior approximation on a simple hierarchical model, and then highlight how calibration changes the decisions of a continue-valued probabilistic matrix factorization model in an intuitive but hard-to-predict manner. Finally, we demonstrate the effect of utility transformations and technical properties of the optimization process. The code for reproducing all experiments (with additional figures) is available online[1].

To highlight the calibration effect, we compare loss-calibrated VI (LCVI) against standard reparameterization VI. The evaluations were carried out for decision problems characterized by loss functions defined for model outputs, i.e., $l(y, h)$ measured by *empirical risk reduction* on test data:

$$\mathscr{I} = \frac{\mathcal{ER}_{\text{VI}} - \mathcal{ER}_{\text{LCVI}}}{\mathcal{ER}_{\text{VI}}}, \quad \mathcal{ER}_{\text{ALG}} = \frac{1}{|\mathcal{D}_{\text{test}}|} \sum_{i \in [\mathcal{D}_{\text{test}}]} \ell(y_i, h_i^{ALG}).$$

Here $\mathcal{ER}_{\text{ALG}}$ denotes *empirical risk* and $h_i^{ALG}$ is the decision obtained for the $i$th point using $\text{ALG} \in \{\text{VI}, \text{LCVI}\}$, optimal w.r.t. loss $\ell$. For $\mathcal{ER}_{\text{VI}}$ we always use the final risk for converged VI, and hence the value of $\mathscr{I}$ corresponds to practical improvement in quality of the final decision problem. On convergence plots, we report its mean $\pm$ standard deviation estimated for 10 different random initializations, and on boxplots the boxes indicate 25th and 75th percentiles.

Whenever not stated differently, we used joint optimization of $\{h\}$ and $\lambda$ with Adam [13] (learning rate set to 0.01) ran until convergence (20k epochs for hierarchical model and 3k epochs for matrix factorization with minibatches of 100 rows). For the first two experiments we set the quantile $M_q$ at 90%, to illustrate robust performance without tuning of hyper-parameters.

### 6.1 Illustration on hierarchical model

The *eight schools model* [9, 29] is a simple Bayesian hierarchical model often used to demonstrate mean-field approximations failing to fit the true posterior. Individual data points are noisy observations $\{(y_j, \sigma_j)\}$ of effects $\theta_j$ with shared prior parameterized by $\mu$ and $\tau$

$$y_j \sim N(\theta_j, \sigma_j^2), \quad \theta_j \sim N(\mu, \tau^2), \quad \mu \sim N(0, 5), \quad \tau \sim \text{half-Cauchy}(0, 5).$$

We couple the model with the tilted loss (Table 1) with $q = 0.2$ to indicate a preference to not overestimate treatments effects, and use the mean-field approximation $q_\lambda(\mu, \tau, \theta_1 \ldots \theta_8) = q_{\lambda_\mu}(\mu) q_{\lambda_\tau}(\tau) \prod_{i=1}^{8} q_{\lambda_{\theta_i}}(\theta_i)$, where each term is a normal distribution parameterized with mean and standard deviation. We used linearized $\mathbb{U}$ with $M$ matching the 90th percentile. Due to small size of the data, empirical risks were calculated on the training data ($\mathcal{D}_{\text{test}} = \mathcal{D}$).

Figure 2 illustrates the calibration process by comparing LCVI with standard mean-field VI and Hamiltonian Monte Carlo using Stan [5], which characterizes the true posterior well and hence provides the ideal baseline. LCVI converges smoothly and provides stable (but small) 1% reduction in risk, validating the procedure. The middle sub-figure illustrates the effect of calibration on the posterior. Standard VI fails to capture the dependencies between $\tau$ and $\theta$ and misses the true posterior mode. Even though LCVI also uses mean-field approximation, it here shifts the approximation towards the region with more probability mass in true posterior. The right sub-figure shows that besides shifting the approximation the calibration process slightly reduces the marginal variances.

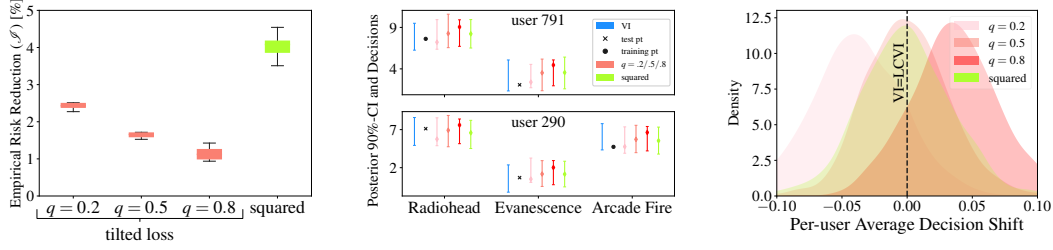

Figure 3: Matrix factorization with tilted and squared losses on the *Last.fm* data set. Loss-calibration clearly reduces the risk (left), while changing the decisions in a non-trivial manner (middle and right).

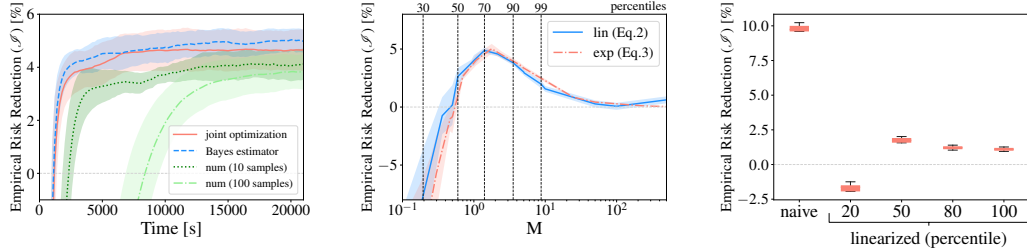

Figure 4: Matrix factorization of *Last.fm* data: (Left:) Joint optimization of $\lambda$ and $\{h\}$ outperforms EM when we need to use numerical optimization of $\{h\}$, but EM with Bayes estimators may be optimal when applicable. (Middle:) The parameter $M$ controlling the transformation of losses into utilities relates naturally to the quantiles of the loss distribution; optimal calibration is here obtained with 70% quantile for both transformations. (Right:) Comparison of different estimators to $\mathbb{U}(\lambda, h)$ in a decision problem expressed in terms of utility.

## 6.2 Matrix factorization and music consumption

We demonstrate LCVI in a prototypical matrix factorization task, modeling the *Last.fm* data set [3], a count matrix $C$ of how many times each user has listened to songs by each artist. We transform the data with $Y = \log(1 + C)$ and restrict the analysis to the top 100 artists. We randomly split the matrix entries into even-sized training and evaluation sets, and provide utilities for predictions.

The effect of loss calibration is best shown on straightforward models with no additional elements to complicate the analysis, and hence we use a simplified probabilistic matrix factorization [19]

$$Y \sim N(ZW, \sigma_y)\, W_{ik} \sim N(0, \sigma_W)\, Z_{kj} \sim N(0, \sigma_z). \tag{8}$$

Here, $Y$ is a $1000 \times 100$-dimensional data matrix, and $Z$ and $W$ are matrices of latent variables with latent dimension $K = 20$. We set all $\sigma$ terms to 10 and use the mean-field approximation $q(\theta) = \prod_k \left[ \prod_i q_{\lambda_{w_{ik}}}(w_{ik}) \prod_j q_{\lambda_{z_{kj}}}(z_{kj}) \right]$, where each term is a normal distribution.

Figure 3 (left) demonstrates the effect of calibrating for squared and tilted loss ($q = 0.2/0.5/0.8$) transformed to utilities by (3) with $M_{90}$ quantile, showing that LCVI achieves risk reduction of $1-4\%$ for all choices. This holds already for the symmetric utilities (squared and tilted with $q = 0.5$) that do not express any preference in favor of over- or underestimation, but simply specify the rate of decline of utilities. The middle sub-figure shows the effect from the perspective of posterior predictive distributions for sample user-artist pairs. The 90%-intervals for all the cases overlap to a high extent, underlining the fact that LCVI calibrates the results of standard VI and usually does not result in drastic changes, and that the decisions behave naturally as a function of $q$; penalizing more for underestimation for larger values. The right sub-figure further explores the effect by plotting the *change* in optimal decision, presented as the the density of user-specific mean differences between LCVI and VI. The individual decisions can change to either direction, indicating that the calibration changes the whole posterior and does not merely shift the decisions.

### 6.3 Algorithm performance

**Optimization algorithm**  Figure 4 (left) compares the alternative optimization algorithms on the matrix factorization problem with squared loss and $M_{90}$ in (3). Here, the EM algorithm with Bayes estimator and joint optimization of $\lambda$ and $\{h\}$ have comparable accuracy and computational cost. However, if we need to resort to numeric optimization of $\{h\}$, EM should not be used: It either becomes clearly too slow (for $S_\theta S_y = 100$) or remains inaccurate (for $S_\theta S_y = 10$). Finally, we note that standard VI is here roughly 10 times faster than LCVI; the calibration comes with increased but manageable computational cost.

**Calibrating with losses**  Section 3.2 explains how losses $l(y, h)$ need to be transfomed into utilties before calibration, and suggests two practical transformations expressed as functions $M$ that can be related to quantiles of the empirical loss distribution of standard VI. Figure 4 (middle) plots the risk reduction for various choices of $M$ to show that: (a) For large $M$ we lose the calibration effect as expected, (b) The optimal calibration is obtained with $M < \sup \ell(y, h)$, and wide range of quantiles between $M_{50}$ and $M_{95}$ provide good calibration, and (c) both the linearized variant and exponential transformation provide similar results. To sum up, the calibration process depends on the parameter $M$, but quantiles of the empirical distribution of losses provides a good basis for setting the value.

**Effect of linearization**  Finally, we demonstrate that linearization may have a detrimental effect compared to directly calibrating for a utility, even for optimal $M$. We use $u(y, h) = e^{-(h-y)^2}$ and compare (6) against (7) (for $\ell(y, h) = 1 - u(y, h)$). Since $\mathbb{U}$ is clearly non-linear close to zero, we see that *direct* calibration for utility clearly outperforms the linearized estimator for all $M$.

## 7  Discussion

To facilitate use of Bayesian models in practical data modeling tasks, we need tools that better solve the real goal of the user. While Bayesian decision theory formally separates the inference process from the eventual decision-making, the unfortunate reality of needing to operate with approximate techniques necessitates tools that integrate the two stages. This is of particular importance for distributional approximations that are typically less accurate than well carried out sampling inference [29], but have advantage in terms of speed and may be easier to integrate into existing data pipelines.

Loss-calibration [16] is a strong basis for achieving this, although it remains largely unexplored. The previous work has been devoted to discrete decisions, and we expanded the scope by providing practical tools for continuous decisions that are considerably more challenging. We demonstrated consistent improvement in expected utility with no complex tuning parameters, which would translate to improved value in real data analysis scenarios. We also demonstrated that for maximal improvement the original decision problem should be expressed in terms of utilities, not losses, in order to avoid detrimental approximations required for coping with decisions based on unbounded losses.

Our work improves the decisions by altering the posterior approximation within a chosen distribution family, and is complementary to directly improving the approximation by richer approximation families [11, 17, 23]. It also relates to the more general research on alternative objectives for variational inference. The research is largely focused on improving the tightness of the bounds (e.g. [6]), but this is not necessarily optimal for all models and tasks [21]. In this context, we provided a practical objective that improves the accuracy in terms of a specific decision task by making the variational bound worse. Finally, recently an alternative approach for improving decisions for approximate posteriors was proposed, based on modifying the decision-making process itself instead of modifying the approximation [15]. The relative quality of these alternative strategies aiming at the common goal of improving decisions under approximate inference is worthy of further research.

### Acknowledgements

The work was supported by Academy of Finland (1266969, 1313125), as well as the Finnish Center for Artificial Intelligence (FCAI), a Flagship of the Academy of Finland. We also thank the Finnish Grid and Cloud Infrastructure (urn:nbn:fi:research-infras-2016072533) for computational resources.

## Footnotes

[1] https://github.com/tkusmierczyk/lcvi

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
