[Supplementary Material 1 · mf_sample_posteriors.pdf]



Figure shows "Posterior 90%-CI and Decisions" for user 791 and user 290 across Radiohead, Evanescence, and Arcade Fire.

Legend:
- VI
- test pt (×)
- training pt (●)
- $q = .2/.5/.8$
- squared

[Supplementary Material 2]



Empirical Risk (top row) and $q\text{Risk}(h^*)$ (bottom row) across columns labeled $q = 0.2$, $q = 0.5$, $q = 0.8$ (tilted loss) and squared.