[Reviews · NeurIPS 2019]

Reviewer 1



Originality: The framework is in general novel. A similar method exists for discrete utilities but not continuous ones. The paper contributes to the loss of calibrated VI for continuous utility. Quality: The work has comparability high-quality. Clarity: In general, the wring is clear. Each step of the work is well-motivated explained. a) However, it would be great to have an explicitly related work section. Due to lack of related work section, related work is either briefly cited such as [14] or not event mentioned, I believe that more related work in approximate inference needs to be mentioned (such as much recent work from Tom Rainforth). b) The linearization in 3.2 should be described to make the paper self-contained. c) The estimation of M_1 using VI should be expaned as well. d) Small things, such as Figure 1 should be put earlier and the caption should be more informative if it is used in the introduction. e) The tasks in the experiments also need more description. Also the recommended one also need to motivate why a continuous utility is needed. For this task, I believe that discrete unity can be used as well. Significance: The technical contribution is incremental but I believe the whole framework can be very useful and significant for many applications.

Reviewer 2



This paper proposes to adapt the variational inference process to better capture the posterior regions relevant to decision-making with continuous utilities (when enumeration is no longer tractable). The result is an importance-weighted ELBO. Using nested Monte Carlo integration and double reparametrization the authors are able to provide the tools to convert continuous unbounded losses into utilities that guarantee optimal calibration. By formulating the problem as a joint optimization procedure of decisions and approximation parameters, the authors showcase how this procedure can be straightforwardly carried out by automatic variational inference. This adds to the significance of his work. Overall, while this paper might lack in originality, it is clear and well written. The proposed tools are sound and were thoroughly studied in terms of calibration. Question: why can we assume smoothness / differentiability of the expected utility ?

Reviewer 3



Originality: The paper builds on ideas developed by Lacoste-Julien et al. (2011) that were introduced to bridge Bayesian decision theory with approximate inference in a meaningful and useful way. The paper takes these ideas and makes them applicable in continuously-valued settings so long as the losses are bounded. For inference, it uses a variation of 'black box' type variational inference schemes. Quality: The paper makes an interesting contribution. However, it is undesirable that the losses must be bounded. Is there a fundamental reason why one cannot extend the proposed methodology to unbounded losses? One potential way of extending/adapting the method that comes to mind is using recent advances in loss-based/PAC/generalized Bayesian posteriors, see e.g. Bissiri, Holmes & Walker, '16 or the work on Generalized VI. The idea there would be to side-step the conversion into utilities altogether. Unless I misunderstood something fundamental, I believe this could be possible by designing a kind of compound loss: adding the negative log likelihood (which is in fact a type of loss itself) to a decision-making-driven loss would generate a new compound/additive loss for the parameter of interest. This in turn would produce an exact & coherent (Gibbs/generalized) posterior in the sense of Bissiri, Holmes & Walker, '16. The variational approximation of such a generalized posterior could then be seen as a form of Generalized VI. If there are fundamental reasons why strategies of this form would be inappropriate/would not be achieving the same goal, it would be good to know what makes them conceptually fundamentally different from utility-based approaches. Clarity: Overall, the paper is well-written and clearly understandable. For me, the one exception to this is the explanation for why one should calibrate the utilities such that their infimum is 0 (lines 107-116). Even after multiple readings, I could not understand where the conclusion that 'for optimal calibration we should use utilities such that inf u(y,h) = 0' comes from. Significance: The paper clearly makes a significant contribution by extending utility/loss calibration into continuously-valued settings. The fact that the loss needs to be bounded however is problematic and should ideally be addressed. EDIT: I was very happy with the reviewer response as it managed to answer my biggest questions convincingly. Accordingly, I am happy to raise my score to a 6 provided that the explanation of the 0-infimum rationale is expanded upon in the main paper in the same way it was expanded upon in the rebuttal -- the extra page will provide the space necessary. Reading your paper once again, I also noticed that using the non-linear loss to utility transformation the authors propose in eq. (3) actually already DIRECTLY corresponds to the kind of compound-loss I was alluding to in my original response. I failed to notice this in my first reading but would encourage the authors to include a short elaboration on this connection in a future version of the paper.

[Author Response · NeurIPS 2019]

We thank the reviewers for the detailed feedback. We will revise the final manuscript to address all the minor remarks,
and answer to the main remarks and direct questions below.

**Reviewer 1**

*a) It would be great to have an explicitly related work section. b) The linearization in 3.2 should be described to make*
*the paper self-contained. c) The estimation of $M_1$ using VI should be expaned as well.*

For the final version we expand the related work by citing recent variational approximation techniques deviating
from direct ELBO optimization [1, 2], make the paper more self-contained by re-iterating the linearization process
(following [14]), and expand the description of the procedure for estimating $M_q$ (which we presume you referred to
with $M_1$). Briefly, we first run standard VI for sufficiently many iterations (until the losses stabilize), and then compute
the loss for every training instance. We then sort the resulting losses and set $M_q$ to match the desired quantile of this
empirical distribution of losses.

*e) Also the recommended one also need to motivate why a continuous utility is needed. For this task, I believe that*
*discrete unity can be used as well.*

Discrete utilities are appropriate for recommender systems, but our demonstration is about predicting consumption
volume, which is better addressed with continuous utility due to very large set of possible values, and because overall
magnitudes matter much more than the exact counts.

**Reviewer 2**

*Why can we assume smoothness / differentiability of the expected utility ?*

We would like to note that smoothness is not a necessary requirement, simply something that helps the optimizer to
converge faster. Based on our experiments and supported by the empirical findings of [23] the expectations of utilities
with point-wise non-differentiability tend to be smooth, even though we do not have a rigorous proof for this.

**Reviewer 3**

*The paper should address if there are ways to use unbounded losses (e.g., by switching from utilities directly to inference*
*in loss-based posteriors). If there is a fundamental reason to use utilities instead, it would be good to have a thorough*
*explanation of this reason.*

The assumptions on utilities (and hence on losses) arise from the derivation of the optimization objective (Eq.1)
that requires the log of the expected utility to be defined. In Section 3.2, we relax these assumptions and provide
two practical ways of handling unbounded losses either by linearization of the utility-dependent term (Eq.7) or by a
non-linear transformation (Eq. 3) that effectively compresses extreme losses into very small (but still non-zero) utilities.
In practice, the procedure seems to work well for unbounded losses as well.

*One potential way of extending/adapting the method that comes to mind is using recent advances in loss-*
*based/PAC/generalized Bayesian posteriors, see e.g. Bissiri, Holmes & Walker, '16 or the work on Generalized*
*VI. I believe this could be possible by designing a kind of compound loss...*

These works provide solid basis for modifying the posterior inference procedure directly and hence offer very
interesting future directions for loss calibration as well. However, it is non-trivial to design such a compound loss
that would directly link the variational objective to the exact definition of gain/risk of Bayesian decision theory. Our
framework achieves this, and we also show (Fig 3c; linearized case) that constructing objectives ad hoc may lead to
suboptimal results. We do not rule out the possibility of alternative loss-calibration strategies building on generalized
posteriors/approximations, and look forward to papers proposing such techniques.

*Why exactly the utility infimum should be 0?*

Optimal decisions $\{h\}$ in Bayesian decision theory remain invariant under linear transformations of $u(.)$. This
does not hold for the loss-calibration bound. Instead, the relative (vs. ELBO) importance of $\mathbb{U}$ is maximal for utilities
(linearly) transformed such that $\inf u = 0$. Whenever $\inf u = \beta$; the bias $\beta > 0$ can be moved outside the integral and
also the logarithm. The remaining term inside the logarithm is $1 + \frac{integral}{\beta}$ that for $\beta \to \infty$ converges to a constant
(e.g., $\nabla \mathbb{U} \to 0$) removing calibration completely. For $\beta \to 0$ the magnitude of the $\mathbb{U}$ term – and hence the calibration
effect – is maximal. We typically want this, but $\beta > 0$ can be used for reducing the calibration effect if so desired.

# References

[1] T. Rainforth et al. Tighter Variational Bounds are Not Necessarily Better. In *ICML*, 2018.

[2] Tao et al. Variational Inference and Model Selection with Generalized Evidence Bounds. In *ICML*, 2018.


[Meta-Review · NeurIPS 2019]

The reviewers have reached a consensus to recommend acceptance. I encourage the authors to take their thoughts into consideration in clarifying the manuscript for the camera-ready version.